# Automated detection of superficial fungal infections from microscopic images through a regional convolutional neural network

**Taehan Koo**[1ᵒ], **Moon Hwan Kim**[2ᵒ], **Mihn-Sook Jue**[1]*

**1** Department of Dermatology, Veterans Health Service Medical Center, Seoul, Republic of Korea, **2** School of Electrical and Electronic Engineering, Yonsei University, Seoul, South Korea

ᵒ These authors contributed equally to this work.
* zooooz@hanmail.net

**Data Availability Statement:** The fungus hyphae data presented in this study are openly available in FigShare at https://doi.org/10.6084/m9.figshare.14678514.v1.

## Abstract

Direct microscopic examination with potassium hydroxide is generally used as a screening method for diagnosing superficial fungal infections. Although this type of examination is faster than other diagnostic methods, it can still be time-consuming to evaluate a complete sample; additionally, it possesses the disadvantage of inconsistent reliability as the accuracy of the reading may differ depending on the performer's skill. This study aims at detecting hyphae more quickly, conveniently, and consistently through deep learning using images obtained from microscopy used in real-world practice. An object detection convolutional neural network, YOLO v4, was trained on microscopy images with magnifications of 100×, 40×, and (100+40)×. The study was conducted at the Department of Dermatology at Veterans Health Service Medical Center, Seoul, Korea between January 1, 2019 and December 31, 2019, using 3,707 images (1,255 images for training, 1,645 images for testing). The average precision was used to evaluate the accuracy of object detection. Precision recall curve analysis was performed for the hyphal location determination, and receiver operating characteristic curve analysis was performed on the image classification. The F1 score, sensitivity, and specificity values were used as measures of the overall performance. The sensitivity and specificity were, respectively, 95.2% and 100% in the 100× data model, and 99% and 86.6% in the 40× data model; the sensitivity and specificity in the combined (100+40)× data model were 93.2% and 89%, respectively. The performance of our model had high sensitivity and specificity, indicating that hyphae can be detected with reliable accuracy. Thus, our deep learning-based autodetection model can detect hyphae in microscopic images obtained from real-world practice. We aim to develop an automatic hyphae detection system that can be utilized in real-world practice through continuous research.

## Introduction

Superficial fungal infections are dermatophyte infections of keratinized tissues, such as skin, hair, and nails. They are among the most common skin diseases with a global prevalence of

**Funding:** The authors received no specific funding for this work.

**Competing interests:** The authors have declared that no competing interests exist.

more than 25%, and the incidence rate is constantly increasing [1]. Clinical detections are helpful in diagnosing superficial fungal infections; however, confirmation through laboratory testing is important for avoiding incorrect diagnosis, unnecessary side effects, and potential drug interactions. Methods currently used to diagnose fungal infections include direct microscopy with potassium hydroxide (KOH) examination, fungal culture, histopathological examination with periodic-acid-Schiff (PAS) staining, immunofluorescence microscopy with calcofluor, and polymerase chain reaction. The KOH examination is generally used as a screening method to diagnose superficial fungal infections because it is relatively convenient, quick, and inexpensive [2, 3]. Through a KOH examination, superficial fungal infections are easily diagnosed under the microscope by their long branch-like structures known as hyphae. To perform a KOH examination of the skin and nails, scales or subungual debris are collected by scraping the involved area with a No. 15 blade. Scraped scales or subungual debris are then placed on a glass slide, prepared with 10% KOH, and capped with a cover glass. When clinicians observe the specimen on a slide under the microscope, they generally screen the entire slide with 40-fold magnification (40×) to find the suspected fungal hyphae region and confirm the hyphae at 100-fold magnification (100×).

Although KOH examination is faster than other diagnostic methods, it still is time-consuming to evaluate a complete sample. Furthermore, KOH examination possesses the disadvantage of inconsistent reliability, i.e., the accuracy of the reading may differ depending on the clinician's skill. In addition, diagnosing multiple samples at once is tedious and can lead to classification errors and increased inter-observer variability. To overcome these conventional limitations, some studies on detecting fungal infections using computer automation techniques are available [3–9]. For example, Mader et al. [6] used multiple image-processing steps to preprocess, segment, and parameterize images obtained using an automated fluorescence imaging system. It is difficult to diagnose fungal infections in real images with conventional computer vision methods because microscopic images contain several other substances apart from dermatophytes. Consequently, previous studies have detected dermatophytes only in images acquired under certain image-processing conditions [6, 8, 9]. To the best of our knowledge, no study has used microscopic images for diagnosing fungal infections.

Therefore, the purpose of this study is to detect hyphae more quickly, conveniently, and consistently through deep learning, a computer automation technology, using images obtained from microscopy used in real-world practice. The deep learning-based autodetection model developed in this study achieved this purpose. Based on our result, we are developing an automatic hyphae detection system that can be utilized in real-world practice through continuous research.

## Materials and methods

This study was conducted in the Department of Dermatology at Veterans Health Service Medical Center and was approved and monitored by the Institutional Review Board (IRB) of Veterans Health Service Medical Center, Seoul, Korea (IRB No. 2020-02-013-001). All image data were obtained from January 1, 2019, to December 31, 2019. Our study did not require patients' personal information, and the IRB approved the exemption of patients' consent.

### Deep learning-based image analysis system

We developed a deep learning-based automatic detection model that detect hyphae in microscopic images obtained from real-world practice through the processes of "sampling and preparation," "data generation," and "test and evaluation" (**Fig 1**).

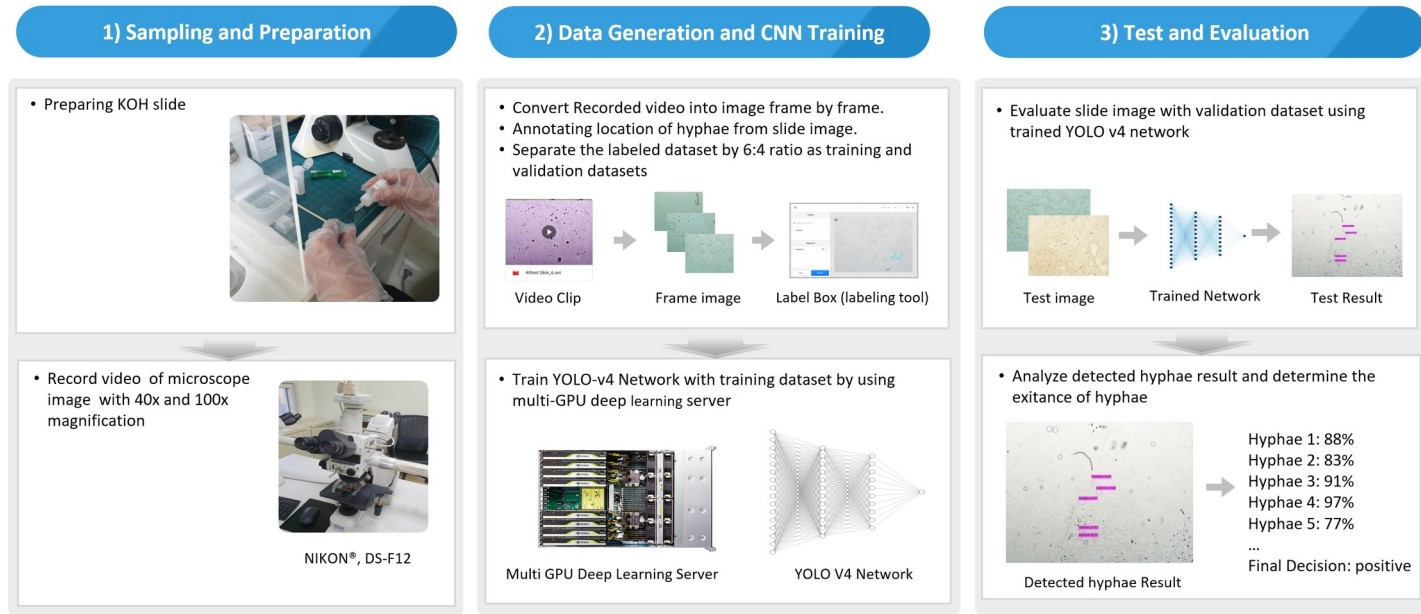

**Fig 1. Workflow for developing a deep learning-based autodetection model that detect hyphae in microscopic images obtained from real-world practice.**

## Sampling and preparation

To perform KOH examination of the skin and nail, scales were collected by scraping the target area outward from the advancing margins with a no. 15 blade. The scraped scales were then placed on a glass slide and covered with a cover slide. Subsequently, several drops of KOH were placed on the slide adjacent to the edge of the cover slide, allowing capillary action to wick the fluid under the cover slide. Two dermatologists read the samples and assigned them to positive and negative classes. The objective of the study is to apply this process in clinical practice by adding a simple device to an existing microscope without expensive equipment such as a digital slide scanner. Therefore, slide images of both magnifications (40× and 100×) were generated as in real-world clinical practice. Image data were acquired through a video captured using a microscope camera (Microscope: NIKON® ECLIPSE E600, Microscope camera: NIKON®, DS-F12), and videos were recorded using a microscope software (iWorks®). To acquire more images of hyphae of various shapes, the images were recorded with 360˚ rotation of the slide where the hyphae were observed. We converted the recorded videos into individual images to label the location of the hyphae.

## Dataset generation

We generated Dataset-100, Dataset-40, and Dataset-all with the captured microscopic images of 100×, 40×, and both 100× and 40×, respectively. In the case of lower magnifications, i.e., Dataset-40, it was possible to observe the overall field quickly. However, the detection accuracy could be low because the observed hyphae are small. At the same time, when observing at higher magnifications, i.e., Dataset-100, multiple scanning jobs were required to check the entire field. However, Dataset-100 has a higher accuracy because the size of the detected hyphae are larger than that in Dataset-40. During both magnifications (40× and 100×), images were constructed and trained on the model. A total of 38 samples were collected from 38 patients, of which 10 positive cases (6 skins, 4 nails), 10 negative cases were at 40× magnification, 8 positive cases (6 skins, 2 nails), and 10 negative cases were at 100× magnification. The

Table 1. Summary of fungus hyphae dataset.

| Dataset | Optical Magnification Ratio (×) | Samples | | | | | | | Image dataset | | | |
|---|---|---|---|---|---|---|---|---|---|---|---|---|
| | | Total | Positive Case | | | | Negative Case | | Total | Positive Case | | Negative Case |
| | | | Training | | Testing | | Testing | | | Training | Testing | Testing |
| | | | Skin | Nail | Skin | Nail | Skin | Nail | | | | |
| Dataset-100 | 100 | 18 | 4 | 1 | 2 | 1 | 5 | 5 | 1279 | 660 | 440 | 179 |
| Dataset-40 | 40 | 20 | 4 | 2 | 2 | 2 | 5 | 5 | 1621 | 595 | 398 | 628 |
| All dataset | 100+40 | 38 | 8 | 3 | 4 | 3 | 10 | 10 | 2900 | 1255 | 838 | 807 |

positive samples were divided into two groups—training and testing datasets (6 training samples and 4 testing samples were at 40× magnification; 5 training samples and 3 testing samples were at 100× magnification). For the positive data (images with hyphae), a practicing dermatologist labeled the location of the hyphae (bounding box) for the entire dataset using "Labeling box" and "YOLO label." We split the labeled image dataset into training and test sets at a 6:4 ratio. As presented in Table 1, each training and testing data images were acquired from the sample obtained in this way. We also created dataset-N (100, 40, all), which included microscopic images without dermatophyte hyphae, for testing. Table 1 summarizes the data used in this study. The fungus hyphae data presented in this study are openly available in FigShare at https://doi.org/10.6084/m9.figshare.14678514.v1.

## Autodetection model using deep learning

The primary objective of automating the KOH examination process is to determine whether a provided microscopy image contains a hyphae object. The following are the two approaches used for determining the image class: image classification and object detection. In the image classification approach, the system determines the class of the provided image as a whole. If the microscopy image contains hyphae, the image classification system returns a positive; if not, it returns a negative. In the object detection approach, the system finds hyphae-like objects and evaluates the similarity of the found objects. If the microscopy image is provided, the object detection system returns the hyphae-like objects with a bounding box that contains the location and size. Thereafter, an additional discriminator determines whether the sample is positive or negative by considering the existence of hyphae-like objects or the probability of hyphae-like objects.

The image classification approach is simpler and more straightforward than object detection. However, it provides only class information: positive or negative. The object detection approach is more sophisticated and complex than the classification approach, and it provides more detailed information about the location and size of the hyphae-like object. Furthermore, differences exist in the databases used by the two approaches. The database for image classification requires only the class of a specific image: positive or negative. However, the database of the object classification approach requires the bounding box information of hyphae objects in the specific microscopic image along with the class. It is more challenging to prepare a database for the object detection system. In this study, the object detection approach was applied. A recently published object detection system, the YOLO v4 network, was applied to obtain a more accurate detection performance.

When a microscopic image is provided, a trained YOLO v4 network analyzes the provided image and outputs each candidate location of hyphae by generating a bounding box within the image. We set a trained YOLO v4 network to extract candidate locations with a reliability of 25% or higher to eliminate insignificant detection results. The intersection over union (IOU)

was used as a cutoff value to determine whether detected locations match with ground truth. If the microscopic image contains a positive object, its IOU will exceed the threshold. The provided image can be described as a bounding box along with its probability. Thereafter, the final decision rule determines whether the microscopic image is positive or negative. When the provided image $I$ has $n$ hyphae objects with probability $P_k$, k = 1,$\cdots$,$n$. we can calculate the minimum, maximum, and average probabilities as follows.

$$P_{max} = \max_{k=1,\cdots,n} P_k \tag{1}$$

$$P_{min} = \min_{k=1,\cdots,n} P_k \tag{2}$$

$$P_{avg} = \frac{1}{n}\sum_{k=1}^{n} P_k \tag{3}$$

In this study, $P_{max}$, $P_{min}$, and $P_{avg}$ were applied as the final probability separately and evaluated as to which value showed the highest performance. If the final probability is greater than the final detection threshold, it is classified as positive. If $n$ is zero or if the final probability is smaller than the final detection threshold, it is classified as negative. The probability threshold proposed in our study is the result obtained by analyzing the ROC value using the MATLAB perfcurve function based on the detection result. The perfcurve function calculates the optimal detection probability threshold by analyzing the ROC curve graph to maximize performance.

## Evaluation

In this study, we evaluated our approach in two ways: (1) evaluating the accuracy of detecting each hyphal object and (2) evaluating the accuracy of the classification results. In the former, average precision (AP) was used to evaluate the object detection problem. The precision recall (PR) curve analysis was performed for the object detection design to determine the hyphal location. F1 scores were derived as a measure to evaluate and compare the overall performance. In the latter, the problem of determining the presence of hyphae in a given microscope slide image is considered a binary classification problem. Thus, a receiver operating characteristic (ROC) curve analysis was performed on the image classification to determine the presence of hyphae in the entire image. In this study, three types of final probabilities ($P_{max}$, $P_{min}$, and $P_{avg}$) were used to determine the final class. To check the difference in each probability, we performed ROC analysis using different $P_{max}$, $P_{min}$, and $P_{avg}$. Finally, we attempted to evaluate and compare the performance of the model to the sensitivity and specificity values of each magnification training type (100×, 40×, and 100×+40×).

## Results

In terms of object detection, performance was obtained through values of the PR curve, F1-score, and AP (**Fig 2**, **Table 1**). In general, the three IOU values of 0.25, 0.5, or 0.75 are used as cutoff values in object detection studies. Our study evaluated the performance of all three values since we first applied the object detection algorithm for hyphae. Our model detects hyphae only in the form of a box, but the hyphae have a characteristic form that cannot be displayed in accordance with the shape of the box because of its curved linear structure. Importantly, to reduce the false-negative rate, the model should be able to detect as many hyphae suspect areas as possible. Therefore, in this study, to increase the search rate, the IOU value was set to the lowest, 0.25. When the IOU was set to 0.25, the recall value of the 100× data model was the highest at 0.93, and the F1-score and AP values were 0.84 and 92.08,

[A] PR Curve for Dataset-40 with IoU=0.25,0.5,0.75

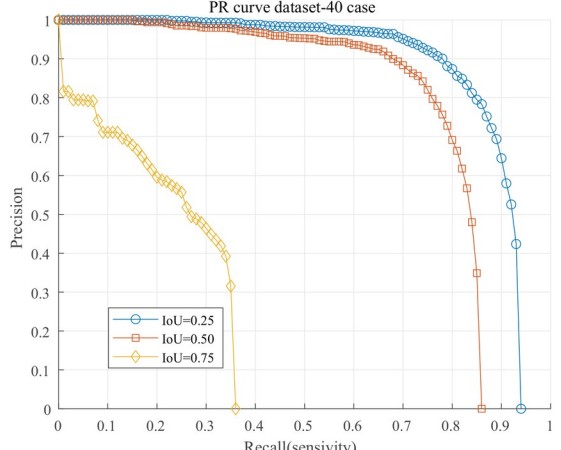

[D] ROC for Dataset-40 with different probabilities

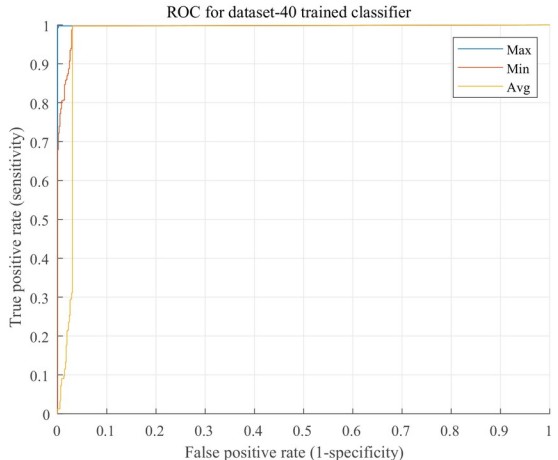

[B] PR Curve for Dataset-100 with IoU=0.25,0.5,0.75

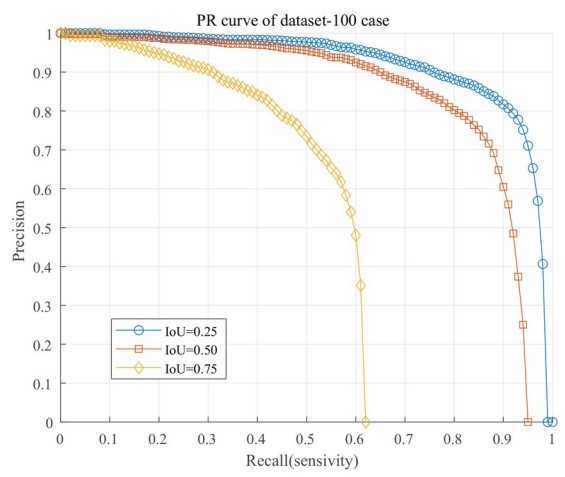

[E] ROC for Dataset-100 with different probabilities

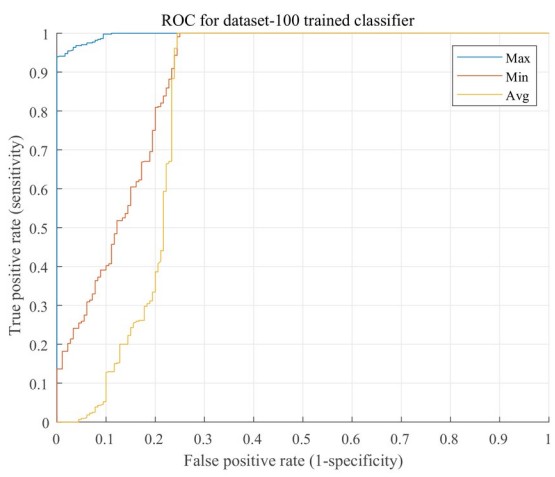

[C] PR Curve for All Dataset with IoU=0.25,0.5,0.75

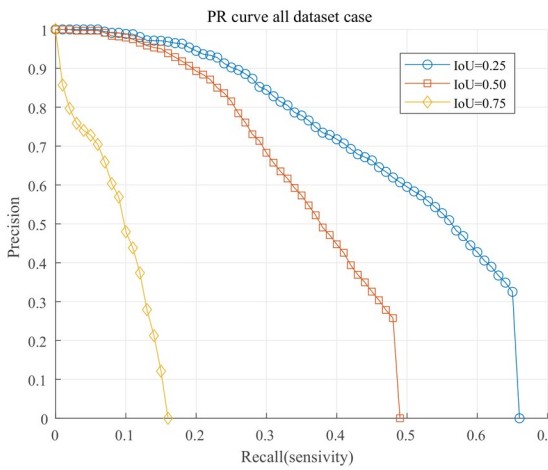

[F] ROC for All Dataset with different probabilities

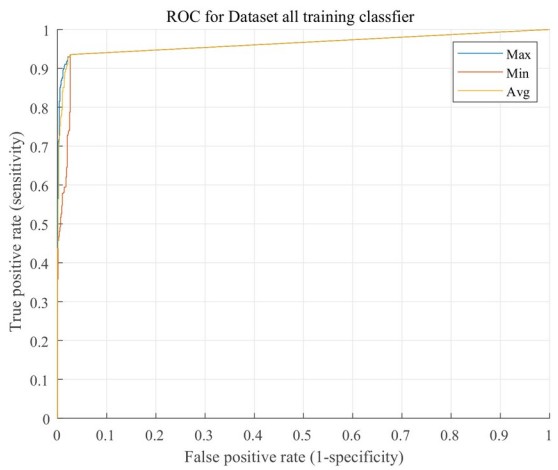

**Fig 2. Precision-recall (PR) curves and receiver operating characteristic (ROC) curves with test datasets.**

**Table 2. Summary of the IOU, TP, FP, FN, precision, recall, F1-score, AP, and AUC values of our model.**

| Dataset | Hyper-detection Accuracy Analysis | | | | | | | | ROC Analysis (IOU = 0.25) | |
|---|---|---|---|---|---|---|---|---|---|---|
| | IOU | TP | FP | FN | Precision | Recall | F1-Score | AP(%) | P_k | AUC |
| **Dataset-100** | 0.25 | 3182 | 970 | 227 | 0.77 | 0.93 | 0.84 | 92.08 | Max | 0.9966 |
| | 0.5 | 2966 | 1186 | 443 | 0.71 | 0.87 | 0.78 | 85.11 | Min | 0.8776 |
| | 0.75 | 2041 | 2111 | 1368 | 0.49 | 0.6 | 0.54 | 52.48 | Avg | 0.8073 |
| **Dataset-40** | 0.25 | 1279 | 256 | 263 | 0.83 | 0.83 | 0.83 | 88.07 | Max | 0.9987 |
| | 0.5 | 1192 | 343 | 350 | 0.78 | 0.77 | 0.77 | 78.8 | Min | 0.9938 |
| | 0.75 | 530 | 1105 | 1012 | 0.35 | 0.34 | 0.34 | 22.24 | Avg | 0.9730 |
| **All Datasets** | 0.25 | 1997 | 799 | 3281 | 0.71 | 0.4 | 0.52 | 50.18 | Max | 0.9650 |
| | 0.5 | 1670 | 11226 | 3281 | 0.6 | 0.34 | 0.43 | 35.97 | Min | 0.9579 |
| | 0.75 | 668 | 2128 | 4283 | 0.24 | 0.13 | 0.17 | 9.23 | Avg | 0.9638 |

IOU: intersection over union, TP: True positives, FP: False positives, FN: False negatives, AP: Average precision, AUC: Area under curve, ROC: Receiver operating characteristics.

respectively. Further, the 40× data model exhibited the excellent performance with the recall, F1-score, and AP values of 0.83, 0.83, and 88.07, respectively. The (100+40)× magnification model exhibited a significantly lower performance than the other two magnification models. The performance of classification, which determines the microscopic image as positive or negative, was evaluated using the ROC curve and area under curve (AUC) values under the setting of the IOU value at 0.25. Between the three types of final probabilities ($P_{max}$, $P_{min}$, and $P_{avg}$), the highest performance was achieved when the final probability was set to $P_{max}$. Consequently, the maximum value of AUC in the 40× data model was the highest at 0.9987, and that in the 100× data model was 0.9966 (**Fig 2**, **Table 2**). The classification performance of all magnification-type models exhibited good performance. The threshold value was calculated as 0.244 by analyzing the ROC curve. We use perfcurve function in MATLAB for ROC curve analyzing. The result of applying the model set in this way to the test data is shown in **Fig 3**. The sensitivity and specificity of the model were respectively 95.2% and 100% in the 100× data model, and 99% and 86.6% in the 40× data model. In the (100+40)× data model, the sensitivity and specificity were 93.2% and 89%, respectively (**Fig 4**).

## Discussion

We developed an autodetection model using deep learning-based computer vision techniques that detect hyphae in microscopic images obtained from real-world practice with high accuracy. Object detection has been an active research area in several fields and aims to determine whether there are any instances of objects from given categories in an image and, if present, to return the spatial location. Recently, deep learning systems have emerged as powerful methods for learning feature representations automatically from data. In particular, these methods have made significant advancements in object detection [3–13]. The object detection technique can be categorized into one-stage and two-stage detectors. The one-stage detector solves image feature extraction and bounding box regression simultaneously, e.g., YOLO [14], SSD [15], and RetinaNet [16]. The two-stage detector is composed of two stages of determining a candidate region based on features and analyzing the bounding box based on the derived region, specifically R-FCN [17], Masked R-CNN [18], and Faster R-CNN [19]. Generally, a one-stage detector has a faster calculation time, whereas a two-stage detector has higher performance. In this study, hyphae objects were detected using the recently published YOLO v4 network, which is a

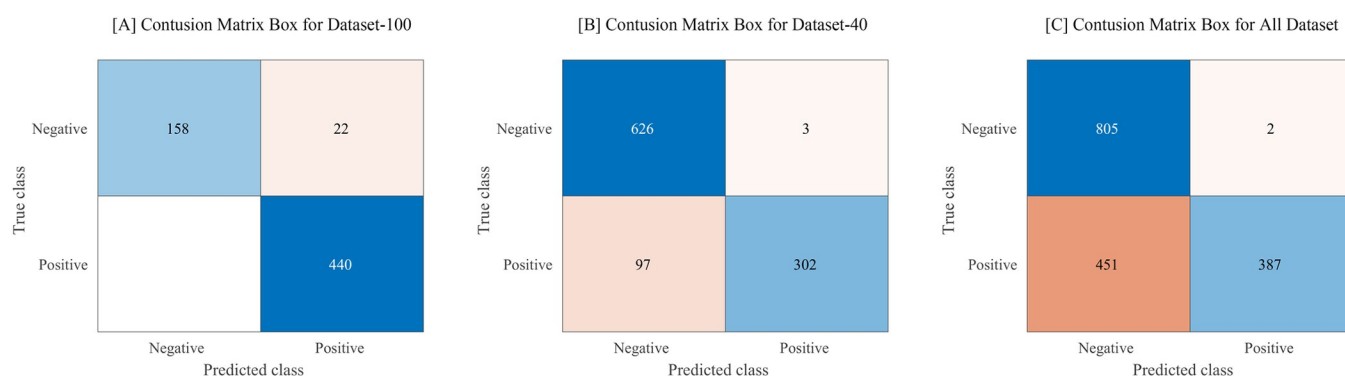

**Fig 3. Example images of the autodetection of hyphae with bounding box.** ¥ (A) positive case with 100× magnification, (B) positive case with 40× magnification, (C) negative case with 100× magnification, and (D) negative case (false detection) with 40× magnification. ¥The ground truth were marked with a green box in positive case (A, B).

**Fig 4.** Confusion matrix box for (A) Dataset-100, (B) Dataset-40, and (C) all datasets.

one-stage detector. This technique possesses several advantages that make it suitable for fungal hyphal detection. First, YOLO v4 is very fast and exhibits higher performance than its previous versions. In particular, its AP and frames per second (FPS) have increased by 10% and 1%, respectively, compared with those of the existing YOLO v3 [14]. Second, YOLO v4 can use a lightweight network structure to apply embedded systems. It is crucial to apply this method to real-world clinical practice. Third, it has sufficient stability because the practicality of the YOLO network has been verified through various applications.

The detection model obtained through our study achieved the sensitivities of 95.2% and 99% in the 100× and 40× data models, respectively. The specificity values of the 100× and 40× datasets were 100% and 86.6%, respectively.

In real-world practice, KOH examination and fungal culture are commonly used to diagnose superficial fungal infections. However, the accuracy of these tests is not as high as expected. In 2010, Jacob et al. reported that the sensitivities for the KOH examination and culture were 73.3% and 41.7%, respectively, and the specificities for those were 42.5% and 77.7%, respectively [20]. The KOH examination has low specificity, and the fungal culture has low sensitivity; thus, accurate diagnosis may be difficult with a single test. Therefore, if our model is applied in real-world medical practice, the diagnosis of superficial fungal infections will be very convenient and have high sensitivity, specificity, and consistent accuracy. Our model provides the classification (positive/negative) of the image, as well as the location and probability of the hyphae object. Accordingly, the clinician can quickly read whether the object boxed by the model are hyphae or not and reduce the entire slide scanning time in the process of KOH examination.

We focused on ensuring that the model has a small false negative value when used as a screening method. If the model returns negative results (no hyphae in the slide), depending on the false negative value, the clinician may have to check all the fields under a microscope to ensure that there are indeed no hyphae. Therefore, it is particularly important that the specificity is high, and the false negative rate is low in the performance of the automatic detection technique system for detecting hyphae. The specificity of our model was high, and the false negative rate was 0% for the 100× data model and 24% for the 40× data model when the IOU value was 0.25.

This study has some limitations. First, our model used 1000 levels of data to obtain results. Although high accuracy has been achieved with 1000 levels of learning, we believe that if more data are collected, more reliable results could be obtained. Second, since our model was developed using image data that was obtained from a setting of single clinic, there is a possibility of performance decrease in different settings of various clinics. The significance of the proposed study is that hyphae can be identified through deep learning techniques. We will continue to train the autodetection model with more data from various clinics. Third, to compare the diagnostic performance of our model with that of experts, we used known expert accuracy. The KOH examination is a commonly used diagnostic method with well-established accuracy that has been previously reported in the literature. This leads us to propose that the accuracies of the autodetection model and the known expert would be comparable.

Although several artificial intelligence (AI) technologies have been studied in connection with medical diagnosis, they are difficult to apply in practice because doctors cannot completely trust the decision made by AI. Our model is valuable in that it attempted an explainable AI approach that provides not only classification, positive or negative, but also object detection: the model finds and displays the location of each hypha as a bounding box. Using our detection model, the doctor could spend more time with patients. Our model has the significant advantage of being able to find hyphae quickly and is reliable owing to its high accuracy. Although heavy multi-GPU machine is required in the training process, a smaller

mobile device is sufficient for the final system that is equipped with the autodetection model obtained through training. A recent study showed that the YOLOv4 model used in our study can be attached to a mobile device [21]. Based on these results, we plan to develop a final system equipped with our autodetection model.

In summary, we developed a deep learning-based autodetection model that detect hyphae in microscopic images obtained from real-world practice. The performance of our model had high sensitivity and specificity, indicating that it is possible to detect hyphae with reliable accuracy. Accordingly, diagnosis can be made more efficiently and in a more straightforward manner. Furthermore, the clinician can quickly check only the hyphae found by the model and confirm it, so that the time spent on microscopic observation can be used for other treatments.

## Author Contributions

**Conceptualization:** Mihn-Sook Jue.

**Data curation:** Taehan Koo, Moon Hwan Kim.

**Investigation:** Taehan Koo, Moon Hwan Kim.

**Methodology:** Moon Hwan Kim.

**Project administration:** Mihn-Sook Jue.

**Software:** Moon Hwan Kim.

**Supervision:** Mihn-Sook Jue.

**Validation:** Moon Hwan Kim.

**Visualization:** Moon Hwan Kim.

**Writing – original draft:** Taehan Koo.

**Writing – review & editing:** Taehan Koo, Moon Hwan Kim, Mihn-Sook Jue.

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
