## [Decision Letter · Decision Letter 0]

8 Mar 2021

PONE-D-21-01387

Automated detection of superficial fungal infections from microscopic images through a regional convolutional neural network

PLOS ONE

Dear Dr. Jue,

Thank you for submitting your manuscript to PLOS ONE. After careful consideration, we feel that it has merit but does not fully meet PLOS ONE’s publication criteria as it currently stands. Therefore, we invite you to submit a revised version of the manuscript that addresses the points raised during the review process.

We look forward to receiving your revised manuscript.

Kind regards,

Mohd Nadhir Ab Wahab, Ph.D.

Academic Editor

PLOS ONE

Journal Requirements:

Additional Editor Comments:

Please address all the comments given by the reviewers before your manuscript can be considered for publication.

Reviewers' comments:

Reviewer's Responses to Questions

**Comments to the Author**

1. Is the manuscript technically sound, and do the data support the conclusions?

Reviewer #1: Yes

Reviewer #2: Yes

2. Has the statistical analysis been performed appropriately and rigorously? 

Reviewer #1: Yes

Reviewer #2: No

3. Have the authors made all data underlying the findings in their manuscript fully available?

Reviewer #1: Yes

Reviewer #2: Yes

4. Is the manuscript presented in an intelligible fashion and written in standard English?

Reviewer #1: Yes

Reviewer #2: Yes

5. Review Comments to the Author

Reviewer #1: This paper presents a novel, deep learning-based approach for detecting fungal infections in microscopic images via the autodetection of long branch-like structures (called hyphae) appearing through KOH examination. The proposed method consists of both location determination of hyphae (object detection) and image classification (positive or negative result for evaluating the presence of infection, respectively). The authors achieved promising results that exhibit rather high sensitivity and specificity, which are claimed to be higher than those of known experts.

In my view the paper provides a valid and valuable contribution towards a more efficient detection of fungal infections, which is certainly not a trivial task. The achieved results are good, nevertheless I see that one of the major limitations that the authors did not dwell into, is that the current results are based on images collected from a single imaging and sample preparation setup and in this way, it's difficult to see whether the proposed model would achieve similarly great results on a more general basis.

Actually, there is a tale-tell sign in the paper regarding this problem, that is, when mixing the datasets with 40x and 100x magnification, there is a significant drop in precision and F1 scores. For this reason I would like to see that that the authors discuss this issue further, particularly in the sense that what could be expected (or what modifications are anticipated in their method) in case they applied the deep-learning based autodetection on a more complete set of images that originate from various imaging setups and settings, as well as taking into account potential variability in sample preparation.

For this very reason I feel that the authors need to be more cautious with a conclusion that the presented results demonstrate a higher sensitivity and specificity than those achieved by manual detection of experts, since the experts can clearly gather practice on a wide variety of setups with differing sample/image quality.

Another points that would require clarification:

(1) The authors use IoU=0.25 value as reference, while it is well-known that an IoU of at least 0.5 is considered a good result in object detection. Therefore I contend that using IoU=0.25 needs some more specific justification in the paper.

(2) Pg. 6, lines 140-141, it is written that: "After calculating the minimum, maximum, and average probabilities, we obtain the final probability for a given

microscopic image." Here and throughout the subsequent evaluation of results it is not clear what 'final probability' means and how the authors arrive at it from the defined min, max and avg probabilities. By the same token, the usage of these probabilities seems rather vague throughout the paper and it does not stand out clearly what is the purpose of these min, max and avg probabilities. Could you please clarify?

(3) Pg. 7. lines 166-167: "The optimal setting of the Fungus hyphae database YOLO v4 model occurs when the IoU value is 0.25 and the detection probability threshold value is 0.244". How do you arrive at the optimal probability threshold 0.244?

A minor note: please provide a proper table legend for Table 2, where all abbreviated measures are clearly given in text (e.g. TP, FP written as true and false positives, which are nowhere else referenced).

Reviewer #2: The authors have recorded a data set of microscopic images for a clinically relevant application case, showing that hyphae detection can be performed well enough for practical purposes using a state-of-the-art object detection network architecture. While there is no methodological advance, I would argue that a well-executed application, along with an annotated data set (that is interesting per se for the image analysis community), should merit a publication.

I have, however, the following comments/requests for revision:

- You mention how many images are contained in the data set, but not how many different skin samples (each of which probably leading to several images). This needs to be clarified. If training and test data contain images taken from the same sample, we would overestimate the performance.

For a real application, the model would need to generalize to completely new samples. Also, you write that samples from both skin and nails were used. Are both distributed equally among training and test data?

- Yolo is a standard object detection network, but by far not the only network suitable for this task. It would clearly strengthen the evaluation if you could report the performance of another method, e.g. a two-stage detector such as Mask R-CNN, as a reference.

- "We also created dataset-N (100, 40, all), which included microscopic images without dermatophyte hyphae, for testing." Does this mean that the negative cases that are available for 40 and 100 are only used during training and not part of the test data?

- "All dataset": Pooling images from different magnifications is not such a practically relevant scenario. In an automated system, all images would be recorded at the same mangnification (or, if different magnifications are supported, one would have separate training sets). I would rather analyze the effect of including/excluding negative cases that will likely occur in a real application.

- The images in Fig.3 differ a lot in their appearance/color. Can you discuss this and maybe provide a few more example images in a supplementary figure? Can you show that positive/negative cases do not differ with respect to their background color, but only with respect to the existence of the target objects?

- Do all hyphae for x100 magnification look like in Fig. 3a and do all hyphae for x40 look like in Fig. 3b? Or is there more variability within x100 and x40, respectively. Again, showing more example images would help.

- Also, does Fig. 3 show full resolution images or can you maybe show them at a higher resolution?

The hyphae in Fig. 3b are really hard to see, and some are actually covered by the labels/boxes: Can this be changed?

- Can you clarify how the data was recorded? What does it mean that "the images were recorded by rotating the screen at a 360 degrees angle where the hyphae were observed"? Why do you record a video instead of still images?

- You claim, already in the abstract, that the network performed better than human experts. In the discussion it then becomes clear that this statement is based on a comparison to values reported in a different context by Jacob et al. Is this really comparable? It is ok to mention their results as a reference, but I would be careful making strong claims in the abstract that are then not fully supported by the results in the paper. In order to make such a claim, you would need an independent ground truth for your data, which would then serve to judge both the network and the human experts.

- The annotated image data set is a valuable contribution of this work and will be of interest for the image analysis community (target objects in front of complex background/also negative examples provided, which is rare). If you plan to publish images and annotations along with the paper (as indicated in the data availability section), I would mention this in the paper, referring to the respective supplementary file/web link.

- Minor points:

- reference [14] contains two papers and should be split

- Fig 4: "contusion matrix": ouch, better write "confusion matrix"

6. PLOS authors have the option to publish the peer review history of their article (what does this mean?). If published, this will include your full peer review and any attached files.

Reviewer #1: No

Reviewer #2: No

---

## [Author Response · Author response to Decision Letter 0]

6 Apr 2021

We greatly appreciate your thoughtful comments that helped us improve the manuscript. 

We trust that all your comments have been addressed accordingly in a revised manuscript. 

We responded to reviewers' comments in as much detail as possible through an attached file named "Response to reviewer". Thank you very much for your effort.

---

## [Decision Letter · Decision Letter 1]

5 May 2021

PONE-D-21-01387R1

Automated detection of superficial fungal infections from microscopic images through a regional convolutional neural network

PLOS ONE

Dear Dr. Jue,

Thank you for submitting your manuscript to PLOS ONE. After careful consideration, we feel that it has merit but does not fully meet PLOS ONE’s publication criteria as it currently stands. Therefore, we invite you to submit a revised version of the manuscript that addresses the points raised during the review process.

We look forward to receiving your revised manuscript.

Kind regards,

Mohd Nadhir Ab Wahab, Ph.D.

Academic Editor

PLOS ONE

Journal Requirements:

Additional Editor Comments (if provided):

Please address all the comments given by the reviewers.

Reviewers' comments:

Reviewer's Responses to Questions

**Comments to the Author**

1. If the authors have adequately addressed your comments raised in a previous round of review and you feel that this manuscript is now acceptable for publication, you may indicate that here to bypass the “Comments to the Author” section, enter your conflict of interest statement in the “Confidential to Editor” section, and submit your "Accept" recommendation.

Reviewer #1: All comments have been addressed

Reviewer #2: (No Response)

2. Is the manuscript technically sound, and do the data support the conclusions?

Reviewer #1: Yes

Reviewer #2: Yes

3. Has the statistical analysis been performed appropriately and rigorously? 

Reviewer #1: Yes

Reviewer #2: N/A

4. Have the authors made all data underlying the findings in their manuscript fully available?

Reviewer #1: Yes

Reviewer #2: Yes

5. Is the manuscript presented in an intelligible fashion and written in standard English?

Reviewer #1: Yes

Reviewer #2: Yes

6. Review Comments to the Author

Reviewer #1: The authors have definitely improved their paper with its revision, raised issues regarding description of the applied methods have been well addressed and clarified.

The only question I still find somewhat ambiguous, and with some claims unwarranted in my view, is the performance of the author's model in terms of (1) comparability to that of dermatologists in real-world practice and (2) whether it can be generalized for other imaging setups than the specific one used in this study. The authors themselves exhibit in their reply to the reviewers' comments that while one could expect that their model may retain good performance in such relations, there is no hard evidence for it at this stage.

Therefore, a cautious approach for deriving conclusions in that regard should be reflected throughout the paper.

Regarding (1), the authors have already softened their claim in the Abstract ("The performance of our model had high sensitivity and specificity, indicating that hyphae can be detected with reliable accuracy."). This, however, is in contrast what is written in the Discussion: line 256-257. : "The performance of our model is higher than the sensitivity and specificity of the known experts, indicating that it is possible to detect hyphae with reliable accuracy. Accordingly, diagnosis can be made more efficiently and in a more straightforward manner." I would suggest that these parts should conform to what is written in the Abstract.

As for (2), I would see it necessary to insert a few sentences again in the Discussion part, where the text would make it clear that: although the applied methodology certainly demonstrates very promising performance, it has been only tested with a single imaging setup and in future work a more thorough testing is needed with a bigger dataset from more variable imaging settings in order to establish how reliable the model's performance is and how it compares to that of real-world dermatologists in such a broader context.

Reviewer #2: The authors have addressed my comments and now provide additional information, figures and access to the data. There are still a few smaller issues that can be resolved by a minor revision:

- Thanks for explaining that you are developing "an automatic hyphae detection system that can be utilized in the field". I would actually mention this in the abstract and make it more clear in the introduction. This would help to introduce your application case to the reader, and it also motivates your decision to prefer a fast network over a potentially more accurate one.

Fig. 1 illustrates the training process with a heavy multi-GPU machine. Maybe something like the image you have appended at the end of the reviewer PDF could be used to illustrate that the final system with the trained model should require only a smaller mobile device?

- I would move the new paragraph between lines 93 and 99 to the section "dataset generation"

You could also include the information about samples and skin/nail into Table 1.

- "We split the labeled dataset into training and test sets at a 6:4 ratio."

I understand this was stratified by sample and skin/nail. Apart from that, was the 6:4 split random?

- Regarding the sentence: "the images were recorded by rotating the screen at a 360 degrees angle where the hyphae were observed"

With you explanations, I now see what you mean. But the sentence is hard to understand and could be improved: You rotate a screen or actually a camera? Rotating by 360 degrees would mean no rotation?

- It would be helpful to also show an example with ground truth annotations. There are some dubious cases where the non-expert reader wonders whether some faint structures are true negatives or actually hyphae that were missed.

7. PLOS authors have the option to publish the peer review history of their article (what does this mean?). If published, this will include your full peer review and any attached files.

Reviewer #1: No

Reviewer #2: No

---

## [Author Response · Author response to Decision Letter 1]

28 May 2021

Response to Reviewer

Dear Edtior,

We thank the reviewers for the insightful comments and suggestions that have helped improve the manuscript significantly. 

We have made several changes to the manuscript according to the suggestions of the reviewers. We appreciate the efforts put in towards the review of this manuscript.

The line numbers mentioned below are based on the “Revised Manuscript with Track Changes” file.

As mentioned earlier (at first revision), we will share the public database link after a legal review once our paper is accepted. As of now, we have made available a part of the image dataset with a label from Figshare, for review purposes—the private Figshare link is https://figshare.com/s/b8f2f80f40b6789daff3. Please note that this private link is only for review purposes, and will be valid for three months. 

Legal reasons for data disclosure can be obtained on the basis of publication, if the manuscript is accepted. We will include the link for the complete dataset once our paper is published.

Reviewer #1: 

The only question I still find somewhat ambiguous, and with some claims unwarranted in my view, is the performance of the author's model in terms of (1) comparability to that of dermatologists in real-world practice and (2) whether it can be generalized for other imaging setups than the specific one used in this study. The authors themselves exhibit in their reply to the reviewers' comments that while one could expect that their model may retain good performance in such relations, there is no hard evidence for it at this stage.

Therefore, a cautious approach for deriving conclusions in that regard should be reflected throughout the paper.

Regarding (1), the authors have already softened their claim in the Abstract ("The performance of our model had high sensitivity and specificity, indicating that hyphae can be detected with reliable accuracy."). This, however, is in contrast what is written in the Discussion: line 256-257. : "The performance of our model is higher than the sensitivity and specificity of the known experts, indicating that it is possible to detect hyphae with reliable accuracy. Accordingly, diagnosis can be made more efficiently and in a more straightforward manner." I would suggest that these parts should conform to what is written in the Abstract.

▶ We understand the concerns raised by the reviewer. The sentence in the ‘discussion’ part (lines 256-257 before) was modified as follows to match the content of the Abstract. Also, lines 232-233 “Clearly, both these indicators (sensitivity and specificity) exhibit higher performances than those of the known expert.” of the Discission were deleted for the same reason.

(Before) Lines 272-273

The performance of our model is higher than the sensitivity and specificity of the known experts, indicating that it is possible to detect hyphae with reliable accuracy. 

(After) Lines 272-273 

The performance of our model had high sensitivity and specificity, indicating that it is possible to detect hyphae with reliable accuracy. 

 

As for (2), I would see it necessary to insert a few sentences again in the Discussion part, where the text would make it clear that: although the applied methodology certainly demonstrates very promising performance, it has been only tested with a single imaging setup and in future work a more thorough testing is needed with a bigger dataset from more variable imaging settings in order to establish how reliable the model's performance is and how it compares to that of real-world dermatologists in such a broader context.

▶ We have added the following sentences in the Discussion section to express the meaning more clearly.

Lines 253-256 

Second, since our model was developed using image data that was obtained from a setting of single clinic, there is a possibility of performance decrease in different settings of various clinics. The significance of the proposed study is that hyphae can be identified through deep learning techniques. We will continue to train the autodetection model with more data from various clinics.

Reviewer #2: The authors have addressed my comments and now provide additional information, figures and access to the data. There are still a few smaller issues that can be resolved by a minor revision:

- Thanks for explaining that you are developing "an automatic hyphae detection system that can be utilized in the field". I would actually mention this in the abstract and make it more clear in the introduction. This would help to introduce your application case to the reader, and it also motivates your decision to prefer a fast network over a potentially more accurate one.

▶ We thank the reviewer for the encouraging words on our manuscript. The following content has been added to the Abstract and Introduction. Further, lines 64-66 of the Introduction have been deleted.

Line 31-33

 We aim to develop an automatic hyphae detection system that can be utilized in real-world practice through continuous research.

Line 66-67

 Based on our result, we are developing an automatic hyphae detection system that can be utilized in real-world practice through continuous research.

-Fig. 1 illustrates the training process with a heavy multi-GPU machine. Maybe something like the image you have appended at the end of the reviewer PDF could be used to illustrate that the final system with the trained model should require only a smaller mobile device?

▶ A heavy multi-GPU machine is required in the training process, but a smaller mobile device is sufficient for the final system that is equipped with the autodetection model obtained through training. A previous study showed that the YOLOv4 model used in our study can be used by attaching it to a mobile device. Figure 1 shows the overall workflow applied in this study. Although our ultimate goal is to develop a small mobile device equipped with a trained model, the results of this study was validated on a multi-GPU machine. Thus, appending the image of a small mobile device in the workflow may cause a misunderstanding among the readers that this paper includes data about the small mobile device, which is not validated in this study. Therefore, we have added the following content to the discussion part of the manuscript and have also added the corresponding reference.

▶ 

Lines 266-269

Although heavy multi-GPU machine is required in the training process,, a smaller mobile device is sufficient for the final system that is equipped with the autodetection model obtained through training. A recent study showed that the YOLOv4 model used in our study can be attached to a mobile device [21]. 

 

- I would move the new paragraph between lines 93 and 99 to the section "dataset generation"

You could also include the information about samples and skin/nail into Table 1.

▶ We thank the reviewer for the insightful comment. The paragraph between lines 97-103 has been moved to lines 112-116 in the section “Dataset generation.” In addition, the skin/nail information has been added to Table 1 as shown below:

▶ Table 1. Summary of fungus hyphae dataset

Dataset Optical Magnification Ratio (×) Samples Image dataset

 Total Positive Case Negative Case Total Positive Case Negative Case

 Training Testing Testing Training Testing Testing 

 Skin Nail Skin Nail Skin Nail 

Dataset-100 100 18 4 1 2 1 5 5 1279 660 440 179

Dataset-40 40 20 4 2 2 2 5 5 1621 595 398 628

All dataset 100+40 38 8 3 4 3 10 10 2900 1255 838 807

-"We split the labeled dataset into training and test sets at a 6:4 ratio."

I understand this was stratified by sample and skin/nail. Apart from that, was the 6:4 split random?

▶ We understand the concern raised by the reviewer. There is no significant difference between hyphae and other floats in skin and nail samples. Therefore, we split the dataset into training and testing sets at a ratio of 6:4 at random without distinction.

- Regarding the sentence: "the images were recorded by rotating the screen at a 360 degrees angle where the hyphae were observed"

With you explanations, I now see what you mean. But the sentence is hard to understand and could be improved: You rotate a screen or actually a camera? Rotating by 360 degrees would mean no rotation?

▶ We understand the concern raised by the reviewer. To express the meaning clearly, the sentence has been modified as follows.

(Before) Line 95-96

the images were recorded by rotating the screen at a 360 degrees angle where the hyphae were observed

(After) Line 95-96

the images were recorded over a 360° rotation of the slide where the hyphae were observed.

 

- It would be helpful to also show an example with ground truth annotations. There are some dubious cases where the non-expert reader wonders whether some faint structures are true negatives or actually hyphae that were missed.

▶ We thank the reviewer for the suggestion. The ground truth annotations have been added to Figure 3. This makes it easier for non-expert readers to compare whether the hyphae observed by the model is real hyphae. 

(Before) Fig.3

(After) Fig.3

Fig 3. Example images of the autodetection of hyphae with bounding box.￥ (A) positive case with 100× magnification, (B) positive case with 40× magnification, (C) negative case with 100× magnification, and (D) negative case (false detection) with 40× magnification.

￥The ground truth were marked with a green box in positive case (A,B).

Additionally, we have made the following revision to the manuscript:

#1. 

We deleted Line 88 “prepared with 10% KOH” because it was duplicated with the following sentence.

#2. 

While moving lines 93-99 to the section “Dataset generation”, the sentence “Two dermatologists read the samples and assigned them to positive and negative classes.” was moved to line 90, and the sentence “As presented in Table 1, each training and testing data images were acquired from the sample obtained in this way” lines 102-103 was deleted.

---

## [Decision Letter · Decision Letter 2]

4 Aug 2021

Automated detection of superficial fungal infections from microscopic images through a regional convolutional neural network

PONE-D-21-01387R2

Dear Dr. Jue,

We’re pleased to inform you that your manuscript has been judged scientifically suitable for publication and will be formally accepted for publication once it meets all outstanding technical requirements.

Kind regards,

Mohd Nadhir Ab Wahab, Ph.D.

Academic Editor

PLOS ONE

Additional Editor Comments (optional):

Reviewers' comments:

Reviewer's Responses to Questions

**Comments to the Author**

1. If the authors have adequately addressed your comments raised in a previous round of review and you feel that this manuscript is now acceptable for publication, you may indicate that here to bypass the “Comments to the Author” section, enter your conflict of interest statement in the “Confidential to Editor” section, and submit your "Accept" recommendation.

Reviewer #1: All comments have been addressed

Reviewer #2: All comments have been addressed

2. Is the manuscript technically sound, and do the data support the conclusions?

Reviewer #1: Yes

Reviewer #2: Yes

3. Has the statistical analysis been performed appropriately and rigorously? 

Reviewer #1: Yes

Reviewer #2: N/A

4. Have the authors made all data underlying the findings in their manuscript fully available?

Reviewer #1: Yes

Reviewer #2: Yes

5. Is the manuscript presented in an intelligible fashion and written in standard English?

Reviewer #1: Yes

Reviewer #2: Yes

6. Review Comments to the Author

Reviewer #1: The authors have addressed all of the concerns raised and I find their answers and modifications to the manuscript satisfactory. The paper in its current form is acceptable for publication in my view.

Reviewer #2: (No Response)

7. PLOS authors have the option to publish the peer review history of their article (what does this mean?). If published, this will include your full peer review and any attached files.

Reviewer #1: No

Reviewer #2: No

---

## [Editor Report · Acceptance letter]

9 Aug 2021

PONE-D-21-01387R2 

Automated detection of superficial fungal infections from microscopic images through a regional convolutional neural network 

Dear Dr. Jue:

I'm pleased to inform you that your manuscript has been deemed suitable for publication in PLOS ONE. Congratulations! Your manuscript is now with our production department. 

Kind regards, 

on behalf of

Dr. Mohd Nadhir Ab Wahab 

Academic Editor

PLOS ONE